# What rainfall rates are most important to wet removal of different aerosol types? Yong Wang<sup>1\*</sup>, Wenwen Xia<sup>1</sup> and Guang J. Zhang<sup>2</sup> <sup>1</sup>Ministry of Education Key Laboratory for Earth System Modeling & Department of Earth System Science, Tsinghua University, Beijing, 100084 China <sup>2</sup>Scripps Institution of Oceanography, La Jolla, CA, USA \*e-mail: yongw@mail.tsinghua.edu.cn

Abstract. Both frequency and intensity of rainfall affect aerosol wet deposition. With a stochastic 23 24 deep convection scheme implemented into two state-of-the-art global climate models (GCMs), a 25 recent study found that aerosol burdens are increased globally by reduced climatological mean wet removal of aerosols due to suppressed light rain. Motivated by their work, a novel approach is 26 developed in this study to detect what rainfall rates are most efficient for wet removal (scavenging 27 amount mode) of different aerosol species in different sizes in GCMs and applied to the National 28 29 Center for Atmospheric Research Community Atmosphere Model version 5 (CAM5) with and without the stochastic convection cases. Results show that in the standard CAM5, no obvious 30 differences in the scavenging amount mode are found among different aerosol types. However, the 31 scavenging amount modes differ in the Aitken, accumulation and coarse modes showing around 32 10-12, 8-9, and 7-8 mm d<sup>-1</sup>, respectively over the tropics. As latitude increases poleward, the 33 scavenging amount mode in each aerosol mode is decreased substantially. The scavenging amount 34 35 mode is generally smaller over land than over ocean. With stochastic convection, the scavenging 36 amount mode for all aerosol species in each mode is systematically increased, which is the most prominent along the Intertropical Convergence Zone exceeding 20 mm d<sup>-1</sup> for small particles. 37 Regardless of whether the stochastic convection scheme is used, convective precipitation has 38 higher efficiency in removing aerosols than large-scale precipitation over the globe even though 39 convection is infrequent over high-latitudes. The scavenging amount modes in the two cases are 40 41 both smaller than individual rainfall rates associated with the most accumulated rain (rainfall amount mode), further implying precipitation frequency is more important than precipitation 42 intensity for aerosol wet removal. The notion of the scavenging amount mode can be applied to 43 other GCMs to better understand the relation between rainfall and aerosol wet scavenging, which 44 45 is important to better simulating aerosols.

#### 47 1. Introduction

Wet deposition through scavenging by rainfall is an important sink for atmospheric aerosols 48 and soluble gases (Atlas and Giam, 1988; Radke et al., 1980). A correlation between the total 49 rainfall amount or rainfall intensity and air pollution has been documented in many studies (Cape 50 et al., 2012; Pye et al., 2009; Tai et al., 2012). For instance, Dawson et al. (2007) found a strong 51 sensitivity of the particulate matter with diameters less than 2.5 µm (PM<sub>2.5</sub>) concentrations to 52 53 rainfall intensity over a large region of the eastern United States from sensitivity tests using a regional numerical model. Besides precipitation intensity, precipitation frequency also influences 54 aerosol wet deposition. In the Geophysical Fluid Dynamics Laboratory (GFDL) chemistry-climate 55 model AM3, Fang et al. (2011) found wet scavenging has a stronger spatial correlation with rainfall 56 frequency than intensity over the United States in January. Mahowald et al. (2011) explored the 57 role of precipitation frequency in dust wet deposition based on model simulations and noted the 58 frequency of precipitation rather than the amount of precipitation controls the fraction of dust wet 59 vs dry deposition outside dust source regions. 60

Hou et al. (2018) investigated the sensitivity of wet scavenging of black carbon (BC) to 61 precipitation intensity and frequency respectively in the Goddard Earth Observing System 62 63 Chemistry (GEOS-Chem) model. The frequency and intensity of precipitation from the GEOS-5 run were used to drive the GEOS-Chem. With the sensitivity tests by artificially perturbating 64 65 precipitation frequency and intensity respectively, they found that the deposition efficiency and hence the lifetime of BC have higher sensitivities to rainfall frequencies than to rainfall intensities. 66 Even with the same mean total rainfall, a different combination of precipitation intensity and 67 frequency results in different removal efficiency of BC. Although these studies investigate the 68 impacts of precipitation intensity and frequency on aerosol wet removal, it is not clear yet 69 70 climatologically what rainfall rates contribute the most to aerosol wet deposition.

Wang et al. (2021) recently found that the frequency of total rainfall in the range from 1 to 20 71 mm d<sup>-1</sup> plays a critical role in regulating the annual mean wet deposition rates of aerosols, 72 especially over the tropics and subtropics. By suppressing the too frequent occurrence of 73 convection in this rainfall intensity range with the introduction of a stochastic deep convection 74 75 scheme (Wang et al., 2016), the aerosol burdens in two global climate models (GCMs) were 76 significantly increased, with the simulated aerosol optical depth (AOD) agreeing better with 77 observations. Based on their work, several interesting questions on the relation between rainfall and aerosol wet removal can be asked: (1) climatologically, what rain rates have the highest 78

efficiency in removing atmospheric aerosols? (2) how much convective and large-scale precipitation contribute to it? (3) for different aerosol types and sizes, does the rain rate most efficient in washing out aerosols differ? (4) also, does it differ over different latitudes and continents/oceans?

To address these questions, this study develops a novel approach to identify the rainfall 83 intensity associated with the most efficient aerosol wet scavenging and applies it to different 84 85 aerosol species at different aerosol sizes in the NCAR CAM5. The paper is organized as follows. Section 2 presents the CAM5 model, experiments, observations and methods. In section 3, 86 precipitation characteristics especially for the amount (defined by daily cumulative rainfall) 87 distributions in two simulations evaluated with observations are presented first. With distinct 88 precipitation features (e.g., frequency and amount) in two simulations, their aerosol wet deposition 89 90 features and mass concentrations are shown. Discussion and conclusions are given in section 4.

91

## 92 2. Experiments, methods and observations

#### 93 2.1. Model and simulations

This study uses the National Center for Atmospheric Research (NCAR) Community 94 95 Atmosphere Model version 5.3 (CAM5.3). As the atmosphere model of the NCAR CESM, CAM5.3 in a standard configuration has a vertical resolution of 30 levels from the surface to 3.6 96 97 hPa and a horizontal resolution of  $1.9^{\circ} \times 2.5^{\circ}$  using finite volume dynamical core. Deep convection is parameterized using the Zhang and McFarlane (1995) (ZM) scheme with dilute 98 convective available potential energy (CAPE) modification by Neale et al. (2008) while the 99 shallow convection scheme uses Park and Bretherton (2009). The Bretherton and Park (2009) 100 moist turbulence parameterization is used to present the stratus-radiation-turbulence interactions. 101 102 The Morrison and Gettelman (2008) (MG) scheme is for large-scale stratiform cloud microphysics. The radiative transfer calculations are based on the Rapid Radiative Transfer Model (RRTM) 103 (Iacono et al., 2008). The properties and process of major aerosol species (sulfate, mineral dust, 104 sea salt, primary organic matter, secondary organic aerosol and black carbon) are treated in the 105 modal aerosol module (MAM) in which distributions of aerosol size are represented by three 106 lognormal modes (MAM3): Aitken, accumulation and coarse modes (Liu et al., 2012). The number 107 108 mixing ratio of each mode and the associated mass mixing ratios of aerosol types in each mode 109 are predicted.

110

Aerosol wet removal consists of in-cloud scavenging and sub-cloud scavenging, which are

111 both treated by the aerosol wet removal module. For in-cloud scavenging, the rainfall production rates and cloud water mixing ratios are used to calculate first-order loss rates for cloud water, 112 which is further multiplied by "solubility factors" to obtain aerosol first-order loss rates. The 113 solubility factors can be interpreted as (tuning factor)  $\times$  (aerosol fraction in cloud droplets). The 114 stratiform in-cloud scavenging only influences the stratiform-cloud-borne aerosol particles with 115 solubility factors of 1.0 (0 for interstitial aerosols). The convective in-cloud scavenging is 116 117 computed both using an in-convection activation fraction and a solubility factor. For stratiform and convective sub-cloud scavenging, the first-order removal rate of interstitial aerosols is equal 118 to (rain rate)  $\times$  (scavenging coefficient)  $\times$  (solubility factor). There is no sub-cloud scavenging 119 for stratiform-cloud-borne aerosols. 120

We use the CAM5.3 simulation output in Wang et al. (2021) for our analysis. The runs with the default ZM scheme (referred to as CAM5) and the stochastic deep convection scheme (referred to as STOC) (Plant and Craig, 2008; Wang et al., 2016) are Atmospheric Model Intercomparison Project (AMIP) type simulations with the present-day (PD) aerosol emission scenario. The prescribed, seasonally varying climatological present-day (averaged over 1982-2001) sea surface temperatures (SSTs) and sea ice extent, recycled yearly force the two simulations which are run for 6 years and the last 5 years are used for analysis.

128

137

## 129 **2.2. Methods**

Both precipitation frequency and intensity contribute to the rainfall amount. Wang et al. (2016, 2021) show that the occurrence frequency of observed and simulated precipitation varies with precipitation intensity largely following exponential functions. Therefore, using a log-linear coordinate system to examine the contribution from each rainfall interval will allow an easier comparison among different rainfall intensity ranges. The contributions from different rainfall rates to the total rainfall amount can be described using the following form (Kooperman et al. 2018):

$$P(R_i) = \frac{1}{\Delta \ln(R)} \frac{1}{N_T} \sum_{k=1}^{N_T} r_k \cdot I\left(R_i^l \le r_k < R_i^r\right)$$

where *i* is the bin index, *r* is the daily rain rate,  $R_i$  is the rainfall bin center with bounds  $R_i^l$  and  $R_i^r$  which is logarithmically spaced covering 4 orders of magnitude of rainfall intensity from 0.1 to 1000 mm d<sup>-1</sup>. The bin width is set to  $\Delta \ln(R) = \Delta R/R = 0.1$ , meaning that the bin interval is 1/10 of the center value (*R*).  $N_T$  is the total number of days, and *I* is a binary operator that has a

(1)

142 value of 1 within the rainfall bin of interest and 0 outside. Thus,  $P(R_i)$  is the contribution to the total precipitation by the rainfall rates centered at  $R_i$ . Graphically, the area under the curve of P in 143 a log-linear plot gives the total amount of mean precipitation. Similarly, within the total 144 precipitation rate bin centered at  $R_i$ , the contributions from convective ( $P_c$ ) and large-scale ( $P_L$ ) 145 146 precipitation are given respectively by:

147 
$$P_{C}(R_{i}) = \frac{1}{\Delta \ln(R)} \frac{1}{N_{T}} \sum_{k=1}^{N_{T}} r_{k}^{C} \cdot I\left(R_{i}^{l} \le r_{k} 

172 individual precipitation intensity most effective in aerosol scavenging is obtained.

The amount distribution of total wet removal of aerosols under different total precipitation intensity can be further decomposed into contributions of wet deposition fluxes from convective and stratiform clouds respectively, similar to the decomposition of precipitation amount:

176 
$$W_{C}(R_{i}) = \frac{1}{\Delta \ln(R)} \frac{1}{N_{T}} \sum_{k=1}^{N_{T}} d_{k}^{C} \cdot I\left(R_{i}^{l} \le r_{k}^{T} 

precipitation in both simulations is overestimated in the tropics and subtropics while that in midand high-latitudes agrees well (Fig. 1a). The overestimated total precipitation over the tropics and
subtropics in both simulations is dominantly from the overestimated convective precipitation (Fig.
1b). Nonetheless, compared to the extremely small large-scale rainfall contribution in the CAM5
run, the increased large-scale precipitation in the STOC run, though mainly contributing to the
further increase of total precipitation in the northern tropics, results in a better agreement with the
TRMM observations.

The distributions of total rainfall amount for GPCP, TRMM, CAM5 and STOC over the 210 tropics (20°S, 20°N), subtropics and midlatitudes (20°N, 50°N), and high-latitudes (50°N, 90°N) 211 are shown in Figure 2a-c. Over the tropics, the distribution in STOC exhibits more rainfall from 212 more intense rain rate and less rainfall from light rain than that in CAM5, thus the rainfall amount 213 mode in STOC (around 40 mm  $d^{-1}$ ) is much stronger than that in CAM5 (~20 mm  $d^{-1}$ ), falling 214 between the TRMM and GPCP observed rainfall amount mode (30-50 mm d<sup>-1</sup>) (Fig. 2a). The weak 215 216 amount mode of total rainfall in CAM5 is controlled by convective precipitation rather than largescale precipitation in terms of their respective distributions and fractional contributions at rain rates 217 ranging from 1 to 20 mm d<sup>-1</sup> (Fig. 2d&g). In contrast, convective and large-scale rainfall in STOC 218 both represents the observed amount mode of total rain. The shift of the total rainfall amount mode 219 to a larger value in STOC is due to the increased (decreased) fractional contribution of convective 220 precipitation at rain rates larger (smaller) than ~20 mm d<sup>-1</sup> (Fig. 2g). Over the subtropics and 221 midlatitudes, the amount mode of total rainfall in CAM5 is comparable to that over the tropics 222  $(\sim 20 \text{ mm d}^{-1})$ . Again, compared with CAM5, the rainfall amount mode in the STOC run shifts 223 rightward better matching GPCP and TRMM observations (Fig. 2b). The representation of 224 convective and large-scale precipitation for the observed amount mode of total rainfall in the two 225 226 simulations is the same as that over the tropics except large-scale precipitation in CAM5 which represents the observed amount mode of total rain as well (Fig. 2e). In contrast to the tropics, the 227 228 difference of the fractional contribution between large-scale and convective precipitation at rain rates between 1 to 20 mm d<sup>-1</sup> in the CAM5 run is reduced due to the decreased convective and 229 increased large-scale fractional contributions (75% vs. 25%) (Fig. 2h). With the introduction of 230 the stochastic deep convection parameterization, the STOC run suppresses the sub-tropical and 231 midlatitude convection, further decreasing their fractional contributions relative to CAM5. At rain 232 rates larger than 20 mm d<sup>-1</sup>, although STOC enhances the fractional contribution of convection, 233 large-scale precipitation, as in CAM5, still makes more contributions. Since large-scale 234

precipitation dominates the total precipitation over high latitudes, the amount distributions of total rainfall are similar between the two simulations (Fig. 2c). Despite this, the amount of convective rainfall and the associated fractional contribution between 1 and 10 mm d<sup>-1</sup> are reduced in the STOC run compared with that in the CAM5 run (Fig. 2f&i).

For a given rain rate, its amount contribution is determined by frequency (f) only  $(P(R_i) =$ 239  $f(R_i)R_i$ ). The frequency distributions of the total precipitation in observations and simulations, 240 and contributions from convective and large-scale precipitation in CAM5 and STOC runs are 241 shown in Figure 3. Over the tropics, where there is frequent convection, although the frequency of 242 total precipitation in the STOC run is slightly higher than that in the CAM5 run at rain rates 243 between 0.1 and 2 mm  $d^{-1}$ , the frequency of rain rates between 2 and 20 mm  $d^{-1}$  in STOC is greatly 244 reduced, much closer to GPCP and TRMM. Furthermore, for rain rates larger than 20 mm d<sup>-1</sup>, the 245 simulated frequency in STOC matches TRMM very well (Fig. 3a). These changes in the total 246 rainfall frequency can be explained by those in individual large-scale and convective components, 247 i.e., a decrease of the frequency of convective precipitation is the main contributor to the frequency 248 change of total rain rates between 2 and 20 mm d<sup>-1</sup> while both large-scale and convective 249 precipitation is responsible for the frequency increase of total rain rates larger than 20 mm d<sup>-1</sup> (Fig. 250 3d). These results are consistent with Wang et al. (2021). As the latitude increases poleward 251 associated with the decreasing frequency contribution of convection, the difference of the 252 253 frequency of total rainfall between CAM5 and STOC runs becomes less prominent (Fig. 3b&c). However, relative to the frequency of convective precipitation in the CAM5 run, similar changes 254 to those over the tropics in the STOC run are still evident (Fig. 3e&f). A chain linking the changes 255 of frequency and amount from CAM5 to STOC is summarized here: with the stochastic deep 256 convection parameterization, the frequency of convection for rain rates between 1 and 20 mm<sup>-1</sup> is 257 258 reduced in STOC, resulting in the decreased amount of total rain within this range and thus the associated shift of the rainfall amount mode to larger rainfall intensity. 259

260

#### 261 **3.2. Wet deposition of aerosols**

With precipitation features in CAM5 and STOC runs in mind, aerosol wet deposition in the two simulations is explored. Figure 4 demonstrates the simulated distributions of wet removal of different aerosol species in different modes over the tropics. Overall, the shape of the distributions of wet removal for all aerosol species in the three modes in both simulations resembles that of the rainfall distribution. Nonetheless, the scavenging amount modes are not equal to the amount modes

of total rainfall as shown in Fig. 2a, especially for large particles. Specifically, in CAM5, for sulfate, 267 sea salt and secondary organic aerosol (SOA) in the Aitken mode, the scavenging amount modes 268 are around 10-12 mm  $d^{-1}$ , smaller than the rainfall amount mode of ~20 mm  $d^{-1}$ . As the aerosol 269 size increases, the scavenging amount modes decrease to 8-9 and 7-8 mm d<sup>-1</sup> in the accumulation 270 and coarse modes, respectively, implying larger particles are easier to be removed by lighter 271 rainfall. The feature that the scavenging amount mode is smaller than the amount mode of total 272 273 rain suggests the frequency of light precipitation plays a more important role in regulating the amount of aerosol wet scavenging than that of rainfall. Additionally, in contrast to other aerosols, 274 the wet removal of sea salt is more sensitive to light precipitation. With the rain rate increasing 275 beyond 1 mm d<sup>-1</sup>, the wet deposition rate of sea salt increases more rapidly than that of other 276 aerosols (i.e., steeper curve). As a response to the shift of the amount mode of total rainfall to a 277 larger value from CAM5 to STOC, the scavenging amount modes for all aerosols in the three 278 modes in STOC are increased accordingly. Owing to the decreased rainfall amount and the high 279 280 occurrence frequency at rain rates smaller than 20 mm d<sup>-1</sup> (Fig. 3a&d), the decrease of wet removal in this rainfall range overwhelms the wet deposition increase at rain rates beyond 20 mm d<sup>-1</sup>. As a 281 result, compared to CAM5, the net decreases of regionally averaged wet removal for all aerosols 282 in the three modes in STOC are found. The largest relative decreases in the Aitken, accumulation 283 and coarse modes are found in black carbon (-33.3% from 0.03 to 0.02 mg/m<sup>2</sup>/day), SOA (-50% 284 285 from 0.004 to 0.002 mg/m<sup>2</sup>/day), and dust (-20.9% from 7.60 to 6.01 mg/m<sup>2</sup>/day), respectively.

The distributions for the subtropics and midlatitudes, and high latitudes are shown in Figures 286 5 and 6, respectively. Same as in the tropics, the similar distributions of different aerosol species 287 in different modes over these two regions are found except for dust in the coarse mode in the 288 subtropics and midlatitudes where two peaks are found: one located at the rain rate around 0.8 mm 289 290  $d^{-1}$  and the other around 8 mm  $d^{-1}$  (Fig. 5). With the suppression of the total rainfall amount between 1-10 mm d<sup>-1</sup> (Fig. 2b), for dust in the coarse mode over (20°N, 50°N), the amount magnitudes of 291 292 two peaks are comparable in the STOC run in contrast to the distinctly different magnitudes of two peaks in the CAM5 run. The scavenging amount modes for all aerosols over these two latitudinal 293 belts are smaller than the rainfall amount modes as well (Fig. 2b&c). In comparison with CAM5, 294 again, the scavenging amount mode shifts rightward and the regional mean of wet removal for all 295 aerosols is reduced in the STOC run (Figs. 5&6). Due to a decrease of mean rain as latitude 296 297 increases, the scavenging amount mode and mean wet removal for all aerosols are increasingly 298 reduced.