# Peer review of "What rainfall rates are most important to wet removal of different aerosol types? Yong Wang1\*, Wenwen Xia1 and Guang J. Zhang2 1Ministry of Education Key Laboratory for Earth System Modeling & Department of Earth System Science, Tsinghua University, Beijing, 100084 China 2Scripps Institution of Oceanography, La Jolla, CA, USA \*e-mail: yongw@mail.tsinghua.edu.cn"

_Atmospheric Chemistry and Physics, 2021_

## Author Comment (AC1)

**Reply to the comments by Reviewer #1**

We thank the reviewer for his/her comments and suggestions on improving our manuscript. These comments are incorporated into the manuscript now. Below is our point-by-point response to these comments. The reviewer's comments are in italic and our responses are in normal font.

*The manuscript mainly studies what rainfall rates are most efficient for wet removal (scavenging amount mode) of different aerosol species in different sizes by using CAM5 with and without the stochastic convection cases. The authors found that larger particles are easier to be removed by lighter rainfall and further suggest the frequency of light precipitation plays a more important role in regulating the amount of aerosol wet scavenging than that of rainfall. Meantime, the authors also pointed out that convective precipitation has higher efficiency in removing aerosols than large-scale precipitation over the globe even though convection is infrequent over high-latitudes. In general, the study is important to understand the relation between rainfall and aerosol wet scavenging. In addition, the paper is well written and presented in a logical way. But, some interpretations and discussions are unclear or missed. I therefore recommend publication of this paper in ACP after major revision. My comments are listed as follows:*

**Reply:** We thank the reviewer for the positive remarks on our work and for the suggestions for further improving the manuscript.

*Major Comments:*

*How to distinguish the convective precipitation and large-scale precipitation? The standard whether is consistent between observation and model?*

**Reply:** In TRMM 3A12 observations, convective and stratiform (i.e., large-scale) precipitation are classified using the brightness temperatures measured by the TRMM Microwave Imager (TMI) radiometer. This is because the local horizontal gradients of brightness temperatures are different in regions with convective and stratiform precipitation. The former is usually characterized by strong gradients of brightness temperature due to large horizontal variations of liquid and ice-phase precipitation, whereas the latter usually has fewer fluctuations of brightness temperature due to relatively weak and uniform updrafts and downdrafts (Kummerow et al. 2001). In global climate models, total precipitation is derived by a process combining resolved grid-scale precipitation explicitly formulated by cloud microphysics schemes (i.e., stratiform or large-scale precipitation generated by the clouds with relatively weak and uniform updrafts and downdrafts) and unresolved sub-grid precipitation formulated by shallow and deep convection schemes (i.e., convective precipitation generated by the clouds with strong updrafts and downdrafts). Although the definitions of convective and large-scale precipitation are not exactly the same between TRMM 3A12 and model simulation, the modeled convective and large-scale (stratiform) precipitation still can be roughly evaluated by using the TRMM 3A12 observations (e.g., Ehsan et al., 2017; Qiu et al., 2019; Chen et al., 2021). We added the description of how the TRMM 3A12 observations derive convective and large-scale precipitation in Lines 232-242 in the revision.

References:

Chen D, Dai A, Hall A. The convective-to-total precipitation ratio and the "drizzling" bias in climate models. Journal of Geophysical Research: Atmospheres, 2021: e2020JD034198.

Ehsan M A, Almazroui M, Yousef A. Impact of different cumulus parameterization schemes in SAUDI-KAU AGCM. Earth Systems and Environment, 2017, 1(1): 3.

Kummerow C, Hong Y, Olson W S, et al. The evolution of the Goddard Profiling Algorithm (GPROF) for rainfall estimation from passive microwave sensors. Journal of Applied Meteorology, 2001, 40(11): 1801-1820.

Qiu L, Im E S, Hur J, et al. Added value of very high resolution climate simulations over South Korea using WRF modeling system. Climate Dynamics, 2020, 54(1): 173-189.

*A main problem of this study is: the author mainly focused on the presentation of physical phenomenon, some important interpretations and discussions are unclear or missed. For example, "why the larger particles are easier to be removed by lighter rainfall?" and "what is the relationship between wet scavenging rates and aerosol types?" The reviewer therefore suggests provide some interpretations and discussions in the result section.*

**Reply:** Thanks for the valuable comments. About "why larger particles are easier to be removed by lighter rainfall", this is because of a combination of higher scavenging coefficients for coarse-mode aerosols in below-cloud scavenging and larger convective-cloud activation fraction prescribed for sea salt and sulfate in the coarse mode according to their hydrophilic properties compared to smaller aerosols. As for "what is the relationship between wet scavenging rates and aerosol types?", generally aerosols with higher hydrophilicity are easier to be washed out. Please see the reply to the comment below for more details. We added interpretations and discussion in Lines 324-327, 331, and 394-396 in the revision.

*What is the difference between in-cloud scavenging and sub-cloud scavenging rate for different aerosol types or precipitation types?*

**Reply:** In CAM5, the aerosol wet removal subroutine treats in-cloud scavenging and below-cloud scavenging. In-cloud scavenging removes cloud-borne aerosol particles (AP) (i.e., aerosols in the cloud droplets) and below-cloud scavenging removes interstitial AP (i.e., aerosols suspended in clear or cloudy air) by precipitation particles through impaction and Brownian diffusion.

For in-cloud scavenging of stratiform clouds, the large-scale precipitation production rates (kg kg$^{-1}$ s$^{-1}$) and cloud water mixing ratios (kg kg$^{-1}$) are used to calculate first-order loss rates (s$^{-1}$) for cloud water (the rate at which cloud-condensate is converted to precipitation within the cloud). These cloud-water first-order loss rates are multiplied by "wet removal adjustment factors" (or tuning factors) to obtain aerosol first-order loss rates, which are applied to activated aerosols within the non-ice cloudy fractions

of a grid cell (i.e., cloudy fractions that contain some cloud water). The stratiform in-cloud scavenging only affects the explicitly treated stratiform-cloud-borne AP which are assumed to not interact with convective clouds, and the adjustment factor of 1.0 is currently used. It does not affect the interstitial AP. In-cloud scavenging in ice clouds (i.e., clouds with no liquid water) is not treated. Cloud-borne particles are treated explicitly and activation is calculated with the parameterization of Abdul-Razzak and Ghan (2000), in which larger and more hydrophilic aerosol particles are easier to nucleate into cloud droplets to form precipitation. The large-scale precipitation production rates, which are generated by cloud microphysics processes, also influence in-cloud scavenging in stratiform clouds.

For convective in-cloud scavenging including shallow and deep convection, cloud fractional area, in-cloud cloud condensate mixing ratio and grid-cell mean convective precipitation production (derived from shallow and deep convection parameterizations) are used to calculate first-order loss rates ($s^{-1}$) for cloud water. Unlike the stratiform cloud-borne AP, the convective cloud-borne AP is not treated explicitly, which is derived by (lumped interstitial aerosols) $\times$ (convective-cloud activation fraction) thus only affecting the grid-cell mean interstitial aerosols. The convective-cloud activation is a prescribed parameter that varies with aerosol mode and species. For example, according to different hydrophilic properties, 0.4 and 0.8 are applied to the dust and sea salt of the coarse mode and a weighted average is applied to the coarse mode sulfate and number. Similarly, these cloud-water first-order loss rates are multiplied by "wet removal adjustment factors" to obtain aerosol first-order loss rates. Here, the wet removal adjustment factor for convective clouds is set to 0.4 to avoid too much wet removal produced by convection.

For below-cloud scavenging of the interstitial aerosol, the first-order removal rate is equal to the product (scavenging coefficient) $\times$ (precipitation rate). The large-scale precipitation rate (from the cloud microphysics scheme) is for stratiform clouds while the convective precipitation rate (from the shallow and deep convective schemes) is for convective clouds. The scavenging coefficient is calculated using the continuous collection equation (e.g., Equation 2 of Wang et al., 2011), in which the rate of collection of a single aerosol particle by a single precipitation particle is integrated over the aerosol and precipitation particle size distributions, at a precipitation rate of 1 mm $h^{-1}$. Collection efficiencies from Slinn (1984) and a Marshall-Palmer precipitation size distribution are assumed. The scavenging coefficient varies strongly with particle size, with the lowest values for the accumulation mode. There is no below-cloud scavenging of stratiform-cloud-borne aerosol.

These details were provided in Lines 131-168 in the revision.

References:

Abdul-Razzak, H. and S. J. Ghan (2000). "A parameterization of aerosol activation 2. Multiple aerosol types." J. Geophysical Research-Atmospheres 105(D5): 6837-6844.

Slinn, W. G. N. (1984). Precipitation scavenging, in Atmospheric Science and Power Production, edited by D. Randerson, pp. 472-477, U. S. Dept. of Energy, Washington D. C.

Wang, X., L. Zhang, and M. D. Moran (2011). "Uncertainty assessment of current size-resolved parameterizations for below-cloud particle scavenging by rain." Atmospheric Chemistry and Physics, 10, 5685-5705. doi:10.5194/acp-10-5685-2010.

*Specific Comments:*

*Line 137: What's the physical meaning of K in the Equ.1? The number of days?*

**Reply:** $K$ is a summation index representing an arbitrary day within $N_T$ days. We defined this in the revision.

*Line 143: Please check the sentence whether is right? "Graphically, the area under the curve of P in a log-linear plot gives the total amount of mean precipitation". Is it total amount of mean precipitation or total contribution?*

**Reply:** Yes, it is correct because $P(R_i)$ in Eq. (2) is the precipitation amount by the rainfall rates centered at $R_i$. We make edits in Lines 184-185 to avoid confusion in the revision.

*Line 178: Where is $d^T$*

**Reply:** We removed it in the revision.

*Figure 1: add the unit of precipitation in the figure or figure caption.*

**Reply:** The unit mm d$^{-1}$ is added in the figure caption in the revised manuscript.

*Figure 2: what's the mean of Y axis in Figure 2? The probability distribution of precipitation amount? Or?*

**Reply:** The Y-axis in the top two rows is the amount of precipitation (i.e., the terms of the left-hand side of Eqs. 2-4) while that in the bottom row is the fractional contribution of convective and large-scale precipitation to the total precipitation.

*The Chen et al., (2017) have compared the dust emissions, transport, and deposition between the Taklimakan Desert and Gobi Desert by using WRF-chem, and found markedly difference exists between these two deserts. My question is: accumulated wet removal of dust whether has regional difference over those Desert regions? Is it totally related with the rainfall rates? What's the role of other factors? Such as, snowfall or hail.*

**Reply:** Thanks for bringing our attention to this paper. In Figures 10 and 11, we can see that over dust source regions such as Sahara, the Taklimakan Desert and Gobi Desert, the rainfall rates associated with 50% of the accumulated wet removal of aerosols are similar in the two simulations both smaller than 2 mm d$^{-1}$. It is because precipitation is scarce over these desert regions, let alone snowfall or hail. Therefore, the dust loadings there are regulated by dust emission, transport and dry deposition. We discussed it and cited this paper in Line 450 in the revision.

*Reference:*

*Chen S. et al. 2017: Comparison of dust emissions, transport, and deposition between the Taklimakan Desert and Gobi Desert. 60 (7), 1338-1355. DOI: 10.1007/s11430-016-9051-0.*

---

## Author Comment (AC2)

**Reply to the comments by Reviewer #2**

We thank the reviewer for his/her comments and suggestions on improving our manuscript. These comments are incorporated into the manuscript now. Below is our point-by-point response to these comments. The reviewer's comments are in italic and our responses are in normal font.

*Summary:*

*In this study, the authors investigate the effects of rainfall frequency and intensity on aerosol wet deposition in versions of CAM5 with the default deep convection scheme and a new stochastic scheme. The authors present an approach to identify the rainfall rates are most efficient for the wet removal of aerosol particles, which is different for Aitken, accumulation and coarse modes, and depends on latitude and land vs ocean. Stochastic convection tends to increase the scavenging amount mode. Of particular interest, is that the rain rates associated with the most scavenging are smaller than the rain rates associated with most rainfall, indicating frequency is more important than intensity for aerosol removal. The reduction in precipitation frequency with the stochastic scheme contributes to higher aerosol concentrations.*

*Overall, the manuscript presents unique research investigating the mechanisms controlling atmospheric scavenging of aerosols by precipitation. It is well motivated and the writing is clear. The methods and results are novel and will be of interest to readers of Atmospheric Chemistry and Physics. This work should be acceptable for publication after revision, which includes additional observational comparison and addressing the major/minor*

**Reply:** We thank the reviewer for the positive remarks on our work and for the suggestions for further improving the manuscript.

*Major Comments:*

*1. The stochastic deep convection scheme should be described in more detail and contrasted with the default ZM scheme. It is introduced on line 122 with no explanation of how it is works and how the reader would expect the results to differ from ZM. I suggest adding a paragraph describing the scheme and providing insights into how the reader might expect it to influence aerosol removal processes.*

**Reply:** Thanks for the comment. A brief description of the stochastic deep convection scheme was added in Lines 92-105 in the revision.

*2. How does the TRMM definition of convective and large-scale precipitation compare to the CAM5 definition of convective and large-scale? Do they mean the same thing in the observations and model? In other words, if you applied the same criteria used to partition TRMM into convective and large-scale components to the partitioning of PRECT from CAM5, would you recreate the same results as PRECC and PRECL? I'm not sure that you would, but this is something that would be worth trying in order to justify the comparison in Figure 1.*

**Reply:** In TRMM 3A12 observations, convective and stratiform (i.e., large-scale) precipitation are classified using the brightness temperatures measured by the TRMM Microwave Imager (TMI) radiometer. This is because the local horizontal gradients of brightness temperatures are different in regions with convective and stratiform precipitation. The former is usually characterized by strong gradients of brightness temperature due to large horizontal variations of liquid and ice-phase precipitation, whereas the latter usually has fewer fluctuations of brightness temperature due to relatively weak and uniform updrafts and downdrafts (Kummerow et al. 2001). In global climate models, total precipitation is derived by a process combining resolved grid-scale precipitation explicitly formulated by cloud microphysics schemes (i.e., stratiform or large-scale precipitation generated by the clouds with relatively weak and uniform updrafts and downdrafts) and unresolved sub-grid precipitation formulated by shallow and deep convection schemes (i.e., convective precipitation generated by the clouds with strong updrafts and downdrafts). Although the definitions of convective and large-scale precipitation are not exactly the same between TRMM 3A12 and model simulation, the modeled convective and large-scale (stratiform) precipitation still can be roughly evaluated by using the TRMM 3A12 observations (e.g., Ehsan et al., 2017; Qiu et al., 2019; Chen et al., 2021). We added the description of how the TRMM 3A12 observations derive convective and large-scale precipitation in Lines 232-242 in the revision.

References:

Chen D, Dai A, Hall A. The convective‐to‐total precipitation ratio and the "drizzling" bias in climate models[J]. Journal of Geophysical Research: Atmospheres, 2021: e2020JD034198.
Ehsan M A, Almazroui M, Yousef A. Impact of different cumulus parameterization schemes in SAUDI-KAU AGCM[J]. Earth Systems and Environment, 2017, 1(1): 3.
Kummerow C, Hong Y, Olson W S, et al. The evolution of the Goddard Profiling Algorithm (GPROF) for rainfall estimation from passive microwave sensors[J]. Journal of Applied Meteorology, 2001, 40(11): 1801-1820.
Qiu L, Im E S, Hur J, et al. Added value of very high resolution climate simulations over South Korea using WRF modeling system[J]. Climate Dynamics, 2020, 54(1): 173-189.

*3. Precipitation observations are presented (GPCP and TRMM), but are there any observational constraints on wet deposition that could be used in this analysis? Reference is given to Wang et al. that has shown that suppressing the too frequent occurrence of rainfall in the light intensity range matches AOD observations, but it would be useful to show that here as well. This would help the reader understand if the increase in concentration with STOC shown in Figure 12 is more (or less) realistic when compared to observations. And is there potentially an observational dataset that specifically assesses wet deposition to take this evaluation further?*

**Reply:** Thanks for the comment. The long-term in situ measurements of aerosol wet deposition by precipitation which can be used in evaluation for climatology are not available. Despite this, for dust wet deposition, a recent study (Kok et al., 2021)

developed an analytical framework that uses inverse modeling to integrate an ensemble of global model simulations with observational constraints on the dust size distribution, extinction efficiency, and regional dust aerosol optical depth. Their inverse dust model agrees better with independent measurements of dust surface concentration and deposition (dry plus wet) flux than the current model simulations and the MERRA-2 dust reanalysis product. Therefore, their gridded dust wet deposition data is used for evaluating dust wet deposition in CAM5 and STOC runs. As seen in Figure R2.1, the annual total amount of dust wet deposition over the globe in CAM5 is 835 Tg, much larger than 702 Tg in Kok et al. (2021). After suppressing too much light rainfall, the value decreases to 646 Tg in STOC, closer to the Kok et al. (2021) value. The comparison of the modeled AOD with MODIS is shown (Fig. R2.2). We can see that the overall performance of modeled AOD in STOC is better than that in CAM5 showing a larger $R^2$ and a smaller RMSE in comparison with MODIS. Figs R2.1 and R2.2 were added as Figures 7&15, respectively, in the revision.

[Figure]

**Figure R2.1.** Global distributions of dust wet deposition in Kok et al. (2021), CAM5 and STOC and the difference between STOC and CAM5. Values are the annual total amount of dust wet deposition over the globe.

[Figure]

**Figure R2.2.** Global distributions of AOD in MODIS, CAM5 and STOC and their differences. The stippled areas indicate that the difference between CAM5 and STOC is statistically significant at the 0.05 level. Values on the top-right corner for the differences between simulations and observations are the coefficient of determination ($R^2$) and the weighted root-mean-square error (RMSE).

References:
Kok, J. F., Adebiyi, A. A., Albani, S., Balkanski, Y., Checa-Garcia, R., Chin, M., Colarco, P. R., Hamilton, D. S., Huang, Y., Ito, A., Klose, M., Leung, D. M., Li, L., Mahowald, N. M., Miller, R. L., Obiso, V., Pérez García-Pando, C., Rocha-Lima, A., Wan, J. S., and Whicker, C. A.: Improved representation of the global dust cycle using observational constraints on dust properties and abundance, Atmos. Chem. Phys., 21, 8127–8167, https://doi.org/10.5194/acp-21-8127-2021, 2021.

*4. In a longterm climatological average, one would expect the sources (emissions) to be about equal to the sinks (wet and dry deposition) of aerosol. Therefore, a difference in total wet removal between CAM5 and STOC would likely be balanced by differences in dry deposition. The burden and lifetime may differ between versions, but if the climate isn't changing over the course of the simulation, the sources and sinks should be in balance. Since aerosol emissions are the same (at least for POM and BC) in CAM5 and STOC, the sinks should be the same. A large difference in total wet removal would imply there is a corresponding difference in dry deposition. Have dry deposition differences been quantified?*

**Reply:** This is a good point. Yes, in general, as wet deposition decreases, dry deposition will increase to make the total sinks unchanged for given anthropogenic aerosol emissions especially for POM and BC. We examined the global averages of dry and wet deposition for different aerosols in CAM5 and STOC simulations (Table R2.1). As you can see, it is true for BC and POM. We discussed this in Lines 358-364 in the revision.

**Table R2.1.** Global averages of dry, wet and total deposition (mg/m$^2$/day) for black carbon (BC), sulfate, dust, sea salt, primary organic matter (POM) and secondary organic aerosol (SOA) in CAM5 and STOC (values in parentheses)

|          | BC       | Sulfate  | Dust     | Sea Salt | POM      | SOA      |
|----------|----------|----------|----------|----------|----------|----------|
| Wet Dep  | 0.0347   | 0.772    | 7.29     | 14.11    | 0.228    | 0.489    |
|          | (0.0344) | (0.797)  | (6.96)   | (14.18)  | (0.226)  | (0.491)  |
| Dry Dep  | 0.0070   | 0.100    | 11.60    | 12.39    | 0.041    | 0.065    |
|          | (0.0072) | (0.111)  | (11.85)  | (13.47)  | (0.043)  | (0.064)  |
| Tot Dep  | 0.0416   | 0.872    | 18.88    | 26.50    | 0.269    | 0.554    |
|          | (0.0416) | (0.908)  | (18.82)  | (27.65)  | (0.269)  | (0.554)  |

*5. In addition to considering the removal of aerosols, more is needed to contextualize the formation of aerosol in the atmosphere. Sulfate is produced interactively in the model through gas-phase and aqueous-phase secondary production, as well as direct emissions. The production of sulfate aerosol is highly dependent on aqueous chemistry in particular, which may be changed with stochastic parameterization and cloud water availability. This can contribute to differences in concentrations and removal as well as the precipitation characteristics. There may be unexplored implications of the*

*intensity and frequency change of precipitation in the stochastic model for aqueous chemistry and resultant removal of sulfate aerosols relative to the production. Are there differences in sulfate production in the two configurations of the model assessed here?*

**Reply:** Thanks for the valuable comment. We investigated the difference of the secondary sulfate aerosol production from aqueous-phase chemical reactions between the two simulations (Fig. R2.3). As seen in Fig. R2.3c, the secondary sulfate aerosol production increases over most regions as a response to the increase of liquid water path (Fig. R2.3d) (Wang and Zhang, 2016). However, when comparing the spatial pattern of changes of secondary sulfate aerosol production with that of the sulfate aerosol burden (Fig. R2.3a), the light-rain frequency change is still the dominant factor contributing to the sulfate aerosol increase (Fig. R2.3a). We have now stated this in Lines 481-484 in the revision.

[Figure]

**Figure R2.3.** Global distributions of differences of (a) sulfate burden (mg m$^{-2}$), (b) light-rain (1<$P$<10 mm d$^{-1}$) frequency (%), (c) secondary sulfate aerosol production from aqueous-phase reactions (μg m$^{-2}$ d$^{-1}$) and (d) liquid water path (g m$^{-2}$) between STOC and CAM5. The stippled areas indicate that the difference between CAM5 and STOC is statistically significant at the 0.05 level.

***Minor Comments:***

*Lines 64: Add a comma: "With the sensitivity tests, by artificially..."*

**Reply:** Done.

*Lines 69-70: Suggested: "it is not clear yet what rainfall rates contribute to the most aerosol wet deposition climatologically."*

**Reply:** Done.

*Lines 79: Suggested: "how much does convective..."*

**Reply:** Done.

*Lines 86: Suggested: "In section 3, precipitation characteristics, especially for amount distributions (defined by daily cumulative rainfall), in two simulations are presented first and evaluated with observations."*

**Reply:** Done.

*Line 135: Suggest adding the Pendergrass and Hartmann, 2014 reference here as well.*

*Pendergrass, A. G., and Hartmann, D. L. (2014). Changes in the Distribution of Rain Frequency and Intensity in Response to Global Warming, Journal of Climate, 27(22), 8372-8383.*

**Reply:** Thanks for bring our attention to this paper. It is cited now in the revision.

*Line 150: Since "aerosol wet scavenging" is not part of equations 1-3, suggest removing this statement here. It should be associated with equations 4-6.*

**Reply:** Done.

*Line 168: "synergy" is not the right word here.*

**Reply:** We changed "synergy" to "combined impacts" in the revision.

*Line 206: Along the lines of the major comment above. Is the partitioning between convective and large-scale in the observations consistent with the partitioning in the model?*

**Reply:** Please see the reply to the major comment #2.

*Line 216: Suggest adding the Kooperman et al. 2018 citation here.*

**Reply:** Done.

*Line 239: It might be better to write this as f = P/R to indicate how f is calculated, since no equation is shown for that earlier.*

**Reply:** Done.

*Line 261: Much of the discussion in this section focuses on the tropics, more attention could be paid to extratropical precipitation as it contributes to the regional nature of precipitation changes between CAM5 and STOC (i.e., Figures 5, 6, 8, and 9 could be described further).*

**Reply:** Thanks for the comment. More discussion regarding Figs. 5, 6, 8 and 9 is added in Lines 343-346, 355-356, and 418-421 in the revision.

*Line 362: As this follows the definition of amount median for precipitation used in Kooperman et al. 2018, suggest citing that here.*

**Reply:** Done.

*Line 420: Suggest changing GCM to CAM5 here. Since no other models are assessed here and this issue is somewhat a result of model formulation, it is not clear if it will be the same in other models with different formulations of wet removal.*

**Reply:** Thanks for the suggestion. GCM is changed to CAM5. More discussion on this is added in Lines 511-523 in the revised manuscript.

*Line 397: PM2.5 was previously mentioned in the introduction as motivation. What are the implications of this research in regard to surface air pollution? Do the authors have some insight as to the nature of the representation of wet deposition on the size distributions relevant to PM2.5? Discussion of this in the conclusion section would help connect to the motivation/introduction.*

**Reply:** As excessive light rain is suppressed, surface air pollution is increased, resulting in some improvements in agreement with observations in China and India (Fig. R2.4). The impact of decreasing light rain on air pollution and its associated health risks are presented in another paper recently also submitted to ACP. Surface $PM_{2.5}$ wet removal is done by below-cloud scavenging. In CAM5, for below-cloud scavenging of interstitial aerosols for both stratiform and convective clouds, the first-order removal rate is equal to the product (scavenging coefficient) × (precipitation rate). The scavenging coefficient is calculated using the continuous collection equation (e.g., Equation 2 of Wang et al., 2011), in which the rate of collection of a single aerosol particle by a single precipitation particle is integrated over the aerosol and precipitation particle size distributions, at a precipitation rate of 1 mm $h^{-1}$. Collection efficiencies from Slinn (1984) and a Marshall-Palmer precipitation size distribution are assumed. The scavenging coefficient varies strongly with particle size, with the lowest values for the accumulation mode. Therefore, for $PM_{2.5}$ particles in the accumulation mode, they will be less efficient to be removed by precipitation than those in the Aitken and coarse modes. We have now discussed this in Lines 524-530 in the revision.

[Figure]

**Figure R2.4.** Observed and simulated surface PM$_{2.5}$ concentrations ($\mu g\ m^{-3}$) over (A) China and (B) India. The inset frame represents the average biases of modeled PM$_{2.5}$ concentrations in each binned observed PM$_{2.5}$ concentrations with an interval of 5 $\mu g\ m^{-3}$.

References:

Slinn, W. G. N. (1984). Precipitation scavenging, in Atmospheric Science and Power Production, edited by D. Randerson, pp. 472-477, U. S. Dept. of Energy, Washington D. C.

Wang, X., L. Zhang, and M. D. Moran (2011). "Uncertainty assessment of current size-resolved parameterizations for below-cloud particle scavenging by rain." Atmospheric Chemistry and Physics, 10, 5685-5705. doi:10.5194/acp-10-5685-2010.